# Investigating the Utility of Surprisal from Large Language Models for Speech Synthesis Prosody

*Sofoklis Kakouros, Juraj Šimko, Martti Vainio, Antti Suni*

University of Helsinki, Finland

{sofoklis.kakouros,juraj.simko,martti.vainio,antti.suni}@helsinki.fi

## Abstract

This paper investigates the use of word surprisal, a measure of the predictability of a word in a given context, as a feature to aid speech synthesis prosody. We explore how word surprisal extracted from large language models (LLMs) correlates with word prominence, a signal-based measure of the salience of a word in a given discourse. We also examine how context length and LLM size affect the results, and how a speech synthesizer conditioned with surprisal values compares with a baseline system. To evaluate these factors, we conducted experiments using a large corpus of English text and LLMs of varying sizes. Our results show that word surprisal and word prominence are moderately correlated, suggesting that they capture related but distinct aspects of language use. We find that length of context and size of the LLM impact the correlations, but not in the direction anticipated, with longer contexts and larger LLMs generally underpredicting prominent words in a nearly linear manner. We demonstrate that, in line with these findings, a speech synthesizer conditioned with surprisal values provides a minimal improvement over the baseline with the results suggesting a limited effect of using surprisal values for eliciting appropriate prominence patterns.

**Index Terms**: prosody, prominence, GPT-2, GPT-J, predictive processing, speech synthesis

## 1. Introduction

While text-to-speech (TTS) has become indistinguishable from human speech for short utterances, it still struggles with contextual prosody. This is not only related to deficiencies in selecting appropriate style, loudness, or emotion, but also in applying appropriate word accentuation patterns based on the linguistic context. The cause of this problem is that the TTS models are largely trained from text-audio pairs of isolated sentences, which limits the quantity and quality of the linguistic data the models are exposed to. On the other hand, the current language models (LMs) are trained to encapsulate the structure, syntax, semantics, and context of natural language, and can then generate responses that are contextually appropriate and coherent.

State-of-the-art LMs can process and generate coherent responses from information that spans much more than a single sentence; a frequent scenario in the case of TTS. Moreover, LMs have been typically trained on vasts amounts of diverse textual data far surpassing the textual data a TTS has been exposed to during training—for instance, GPT-2 has been trained on the WebText dataset that consists of 40Gb of text. Current state-of-the-art large language models (LLMs) such as GPT-2 [1] can process very large contexts that are several orders of magnitude larger than the typical data used in TTS. GPT-2 can manage contexts up to 1024 tokens, GPT-3 [2] 2048 tokens,

while more recent models such as GPT-3.5 can process up to 4096 tokens [3] and GPT-4 8192 with a capability of up to 32768 tokens [4]. The obvious question arises, how can we leverage LLMs in improving the contextual prosodic appropriateness of TTS systems?

One approach is through analogy. Language models are very good learners of the statistical structure of the language. The models learn to predict the most likely sequence of tokens given an input that is constrained within a contextual window. Similarly, as humans, we have the remarkable ability to comprehend and anticipate the flow of written or spoken communication. When we read a passage, we are able to identify the general direction of the author's thoughts and ideas. Although we cannot precisely predict every word in the flow of text that the author has written, we can identify sudden shifts in topic or language that do not align with the context.

These type of shifts can be seen as surprising by humans and similarly, when given to an LM, an incongruous input is interpreted as a low likelihood transition in a sequence of words. Surprisal has been shown in the literature to be connected to the impression of highlighting, and in prosodic terms, to the phenomenon of prosodic prominence [5]. In general, low likelihood textual or acoustic input has been shown to correlate well with the subjective impression of prominence in speech [6, 7, 8]. However, LMs or LLMs have not been used before for the automatic annotation of prominent words in speech based on their surprisal values.

In this work, we use the GPT-2 family of models and GPT-J (an open-source and open-access alternative to GPT-3) for the estimation of the surprisal values based on textual materials from the LJ Speech corpus [9]. With the computed surprisal values and for different context lengths, we analyze the relationship between surprisal, context, and prominence. We then train a TTS system with FastSpeech [10] adding the surprisal values in the training of the LJ Speech corpus. Our method is evaluated with objective metrics between the original and generated speech signals.

### 1.1. Prosody

The prosody of speech is important for the correct interpretation of language. Prosody comprises of a range of phenomena that affect the perceived naturalness of speech; how something is said rather than what is said. This is typically encoded in speech through variations in pitch, rhythm, and timing [11]. Although prosody is largely an aspect of spoken language, prosodic structure is also partly linguistically encoded in text. This has enabled the use of LMs and LLMs for the detection of prominent words from textual resources. For example a recent study showed that indeed LMs, such as BERT, can be used for promi-

nence detection when fine-tuned on a corpus with prominence labels [12]. In another study it was shown that BERT can leverage the structural information of the language it has learned during its pre-training that encodes surface linguistic features such as part-of-speech (POS) tags and semantic information of the language for the detection of prominence [13].

In general, prosodic phenomena such as prominence arise from local or long-term contextual dependencies that can be potentially captured by LMs. As prosody can span individual units, such as words, and is dependent on larger contexts that may involve one or more sentences or even paragraphs, the capacity of LMs that can process large contextual windows becomes increasingly relevant for the study of prosody.

### 1.2. Language models

Language modeling is a natural language processing (NLP) task that involves predicting the likelihood of a sequence of words in a language. Language models (LMs) are statistical learners that can capture the general context of text and model the relationship between subsequent tokens. The state-of-the-art methods for language modeling have evolved from traditional LMs, such as $n$-grams, to large language models (LLMs), such as GPT-2, which can capture more context and model large contextual dependencies that span many words, sentences, and even paragraphs. LLMs have shown significant improvements in performance compared to smaller language models, especially on tasks that require a deeper understanding of language.

The GPT (Generative Pre-trained Transformer) family of language models [1] together with BERT (Bidirectional Encoder Representations from Transformers) [14] represent the most popular models publicly available. GPT-2, GPT-3, and more recently GPT-3.5 (ChatGPT is a model in the GPT-3.5 series) and GPT-4, are all autoregressive transformer-based models that generate text based on the sequence of words already generated. This means that the GPT models have a unidirectional attention flow that enables them to process the past context only when predicting the next word. This is in contrast to BERT that has bidirectional flow allowing the model to use context from both directions during processing.

### 1.3. Predictive processing

Predictions play a crucial role in our ability to plan forward and prepare our actions. Several investigations have provided evidence that our brain represents sensory information probabilistically [15, 16]. Predictive processing is a theoretical framework that has been used to explain how humans perceive and process information. This framework suggests that the brain uses predictions to interpret sensory input, which helps to reduce the amount of processing required to make sense of the world [15]. Predictive processing has been applied to various domains, including language processing and prosody [17, 18].

As described in [18], every linguistic stimulus we process comes with a context. Respectively, on the basis of the previous linguistic contextual information, a stimulus, can be deemed to be more or less expected. On this basis, we propose that LMs can be used to process linguistic information of different context lengths and determine surprisal based on the probabilities the models find for each word token to be given their past context.

## 2. Related work

The application of predictive-based theories in connection with cognitive modeling is an idea that has been around for long

and applied in many areas including language perception and production [19]. Our motivation for this study draws partly from the work in [19] who investigated the processing difficulty (measured in terms of reading times) with respect to the degree of predictability of upcoming linguistic content given its context and showed that the predictive power increased as a linear function of the language model's size. However, their work was based on simple models involving $n$-grams and LSTMs, whereas a recent work [20] showed that using newer models, such as GPT-2, underpredicts reading times, and that the degree of underprediction increased with model size. Similar to these works, our aim is to use a predictive-based approach that quantifies the degree of surprisal of individual words given their context, but instead of looking at reading times, our investigation focuses on the connection of surprisal with prosodic prominence.

When it comes to modeling the prosody in TTS, there is a history in using language model statistics for prosody prediction, mainly as features for predicting pitch accent from text. However, majority of these works make use of small language models such as unigrams and bigrams [21]. More recently, prediction of prominence using BERT [12] has provided promising results but longer contexts have not been taken into account in the design. BERT embeddings have also been sometimes incorporated into TTS [22]. Contrary to these previous studies, in our current work, we do not estimate prominence explicitly for TTS but evaluate the efficacy of using the continuous surprisal values directly in TTS training aiming to improve the prosody of the generated speech.

## 3. Method

### 3.1. Data

For our experiments we use the LJ Speech dataset [9]. LJ Speech is a public domain speech dataset that consists of 13,100 high-quality speech recordings of a single speaker reading passages from 7 non-fiction books, collected from Librivox. The recordings are from a female speaker of US English, and each recording varies in length from 1 to 10 seconds. The total duration of the dataset is approximately 24 hours. The reader is a professional, with lively, yet somewhat idiosyncratic prosody. Importantly, the reading is continuous, allowing for examining the effect of context in this study.

### 3.2. Word surprisal

To compute word surprisal we use the predictions from four variants of GPT-2 and the GPT-J model (all models are hosted in the `Hugging Face` model library)—GPT-2 models are trained on 40Gb of texts (WebText dataset) and GPT-J on 825Gb (the Pile dataset). In particular, we use GPT-2 small (`gpt2`; 124M parameters), medium (`gpt2-medium`; 355M), large (`gpt2-large`; 774M), and extra large (`gpt2-xl`; 1.5B). For GPT-J, we use the `EleutherAI/gpt-j-6b` model that contains 6B parameters and is comparable in performance to GPT-3. For unigram computation we used publicly available word counts derived from the `Google Web Trillion Word Corpus`[1].

The GPT family of models process a given text sequence using tokens. Tokens are common sequences of characters found in text. This means that a single word will not necessarily find a match in the model's dictionary and it might be split into a sequence of tokens. Tokenization is performed in order

---

[1]https://norvig.com/ngrams/count_1w.txt

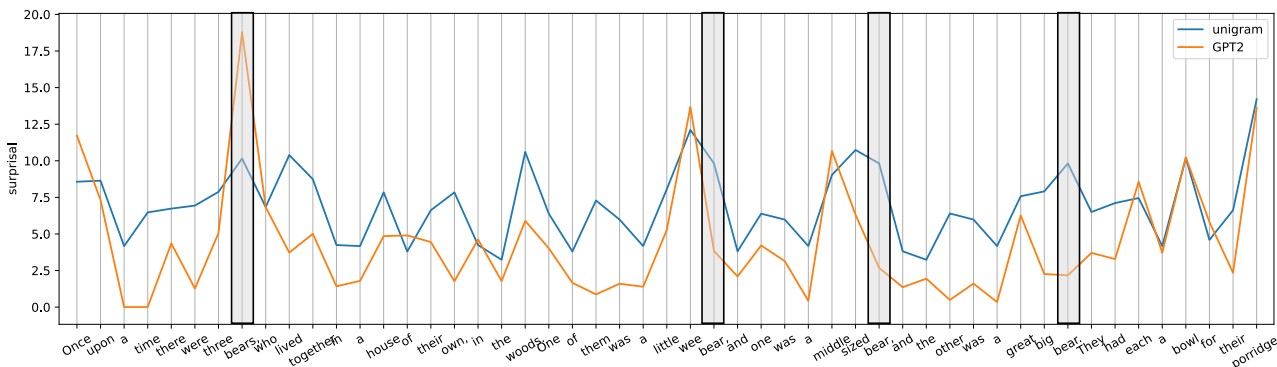

Figure 1: *Context-dependence of GPT surprisal values compared with unigrams.*

to reduce the model's dictionary size and to enable handling of out-of-vocabulary (OOV) words. The GPT models learn the statistical relationships between these tokens, and produce the next token in a sequence of tokens. Word surprisal can be seen as an information-theoretic measure of the amount of new information conveyed by a word [23, 19]. In our experiments, we extract word surprisal values by taking the aggregate token surprisal over each word as follows:

$$\mathbf{w}_t = \sum_{\tau=0}^{N} \mathbf{S}_{\tau,L}, \qquad (1)$$

$$\mathbf{S}_{\tau,L} = -log_2 \mathbf{P}(w_\tau | w_{\tau-1,\dots,\tau-L}), \qquad (2)$$

where $\mathbf{S}_{\tau,L}$ is the surprisal value for token $\tau$ given $L$ previous tokens of word $t$ ($\mathbf{w}_t$) that consists of $N$ tokens.

In some cases in our analysis we also make a distinction between the surprisal of *stop-words* and *content* words. These word classes were extracted using spaCy[2] (an NLP library in Python) where *stop-words* represent words that appear very commonly in the language such as 'I' or 'and' whereas *content* words are words that convey semantic content and contribute to the meaning of the sentence (e.g., nouns, verbs, adjectives).

### 3.3. Context modeling

To model larger textual contexts than a single sentence, we construct a context for each target sentence by prepending the target sentence with the text segments that preceded it. In case the sentence was at the beginning of the chapter, the context was left empty. For our analysis, we included context of up to 5 previous segments —context sizes from $0-5$ are denoted in the text as sup_0, sup_1, sup_2, sup_3, sup_4, and sup_5 where $0$ refers to a sentence without context. Note that a segment is an entire transcription of an LJ Speech recording that appears in the correct order in the running text.

### 3.4. Prominence estimation with CWT

In order to assess the correlation of surprisal values with speech prosody, we utilized word prominence estimates, derived automatically using Wavelet prosody toolkit[3]. This prominence estimation method combines $f_0$, energy and duration information into a composite signal, and performs a continuous wavelet transform (CWT) on the signal and integrates peaks over selected time scales. The method yields prominence estimates

that have been shown to correlate well with perceptual prominence in English [24]. Unlike in some previous studies [25, 12], we did not quantize the prominence values, as the continuous estimates are more appropriate in comparison with similarly continuous surprisal values.

### 3.5. Speech synthesis

For studying the effects of word surprisal on speech synthesis prosody, we applied a transformer-based FastPitch[26, 27] model architecture augmented with local conditioning. FastPitch has desirable features for the current study, including explicit modelling of segment level prosodic features, $f_0$, intensity and duration. Additionally, it incorporates a robust alignment framework [28], leading to fast convergence and thus allowing for quick experimentation. To implement the local conditioning, we repeated the continuous conditioning features (word prominence or surprisal value) for each segmental sound (phone) of the word. We then embedded the features using a linear layer to a dimension of 384 and summed the resulting embeddings with the phone representations of the FastPitch encoder.

## 4. Comparing surprisals, context and prosody

### 4.1. Givenness

Given words and concepts, i.e., the words and concepts already previously introduced in a narrative, can be expected to be realized with less prominence than the novel ones [29, 30, 31]. Unlike unigrams, surprisal-based models operating on a large context can be expected to account for this type of givenness, attributing higher probability (lower surprisal) to the words that had occurred previously in the text, particularly for content words. Fig. 1 illustrates this clearly: the word 'bear' is assigned progressively lower surprisal values by the GPT2 model, the trend, obviously, not depicted by the unigram surprisal. As we can expect that this word will be realized with decreasing level of prominence as the sentence goes on, the context-aware surprisal values models should help elicit appropriate differences in prominence in speech synthesis implementation.

In order to compare the givenness-related prominence patterns with corresponding surprisal values within the analysed corpus, we assigned each content word in the corpus a distance from its previous occurrence in the following way: if the same word occurred previously in the same sentence, the assigned value is 0, if it previously occurred in the previous one, the value is 1, if in the one before, the value is 2, etc.

[2] https://spacy.io/ (spaCy 3.5.2)

[3] https://github.com/asuni/wavelet_prosody_toolkit/

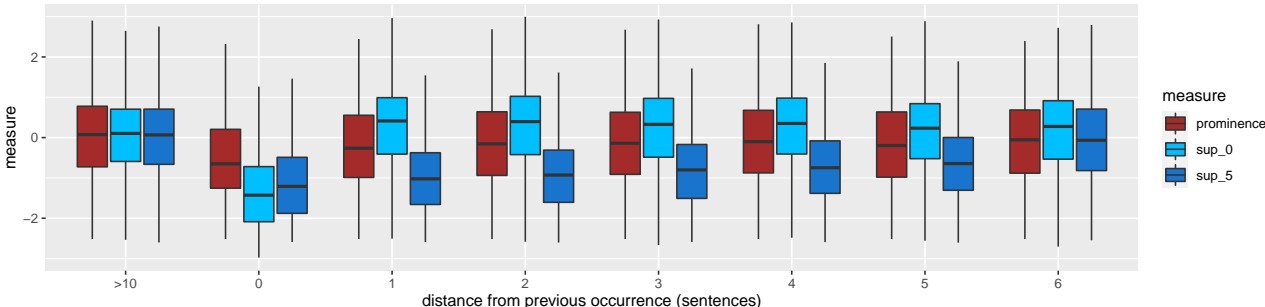

Figure 2: *Dependence of* `prominence`, `sup_0` *and* `sup_5` *values on givenness.*

Fig. 2 shows the dependence of `prominence`, and surprisal values `sup_0` and `sup_5` from GPT-J model on this distance measure for content (non stop-words) in the corpus. In order to facilitate comparisons, all values were normalized with respect to "novel" values (with distance greater than 10 sentences) by subtracting the mean and dividing by the standard deviation of the "novel" values.

As seen in the figure, the prominence level on average decreases for the words repeated within the same sentence (distance value 0); for the occurrences with mention in the previous few sentences, the prominence level is (on average) somewhat lower than for 'novel' words (distance > 10), but to a considerably smaller degree. The GPT model with longer context (`sup_5`) behaves similarly, but greatly exaggerates the context dependency: the surprisal levels remain considerably lower for all words repeated within the scope of the model (last 5 sentences). For the short-context model (`sup_0`), the occurrence of the word given in the previous sentence naturally does not influence surprisal; in fact, the surprisal values are higher for the repeated words, probably because the content words that are less predictable than average might be more likely to be locally repeated in the narrative.

### 4.2. Correlations between signal-based prosodic characteristics and surprisals

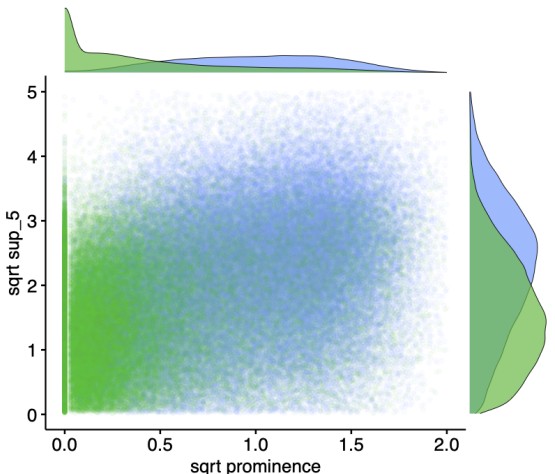

Figure 3: *Scatter plot of* `prominence` *values against corresponding* `sup_5` *values generated by GPT-J model for all words in the corpus, separately for stop-words (in green) and non stop-words (in blue). Density plots for the words group are plotted against the respective axes. Square roots of both measures were used for plotting.*

In this section we proceed with comparing the various surprisal measures with signal-based prosodic characteristics, including `prominence` estimated using CWT method.

Fig. 3 depicts the relationship between word-level continuous prominence values and the corresponding surprisal values calculated using the GPT-J model with long previous textual context (5 previous sentences). In general, both `prominence` and `sup_5` for stop-word are smaller than those for non stopwords (content words), indicating a degree of bimodality in the distributions based on the word category. That means that potential correlation between these two measures that can be used by speech synthesis system to elicit appropriate degree of prominence based on surprisal value might be limited to this bimodality.

Beyond that, the scatter plot does not reveal any clear pattern of interdependency of these two measures, in particular for the content words (a weak relationship somewhat visible for stop-words is largely obscured by the content word points in the centre of the figure). In order to evaluate this observation for multiple prosodic features (in addition to `prominence`) as well for several tested language models and context sizes, we calculated correlations between respective word-level surprisal values and corresponding signal-based characteristics.

Fig. 4 plots the resulting Spearman's rank correlations. The surprisal values include those calculated from unigrams, as well as from the models using previous textual context of varying length (`sup_0`–`sup_5`). The correlations were calculated for all words in the corpus (in red in Fig. 4) as well as separately for stop-words (green) and content words (blue), respectively.

Generally, the correlations are comparatively higher for the entire corpus than for the two sub-groups separately (and higher for stop-words than for content words), in line with the observed bimodality of the compared distributions. Also, rather interestingly, the correlations are generally *higher* for the smaller models than for the larger ones, and, in most cases, decrease with the size of the context used for the surprisal values. The measures of `prominence`, `duration` and $f_0$-`sd` correlate best with unigram-based surprisal. The sole exception to this pattern are the relatively small correlations between `prominence` and surprisal values for content words, where correlation *increases* with longer context length and model size. For `intensity-sd` measure, the correlations are greatest for `sup_0` model, and for the sub-groups remain higher for GPT-based language models than for unigrams, particularly for stopwords (for this category, the larger models yield higher correlations than smaller ones).

The correlation plots for $f_0$-`mean` and `intensity-mean` measures reveal different patterns. While the correlations are considerably lower than for most

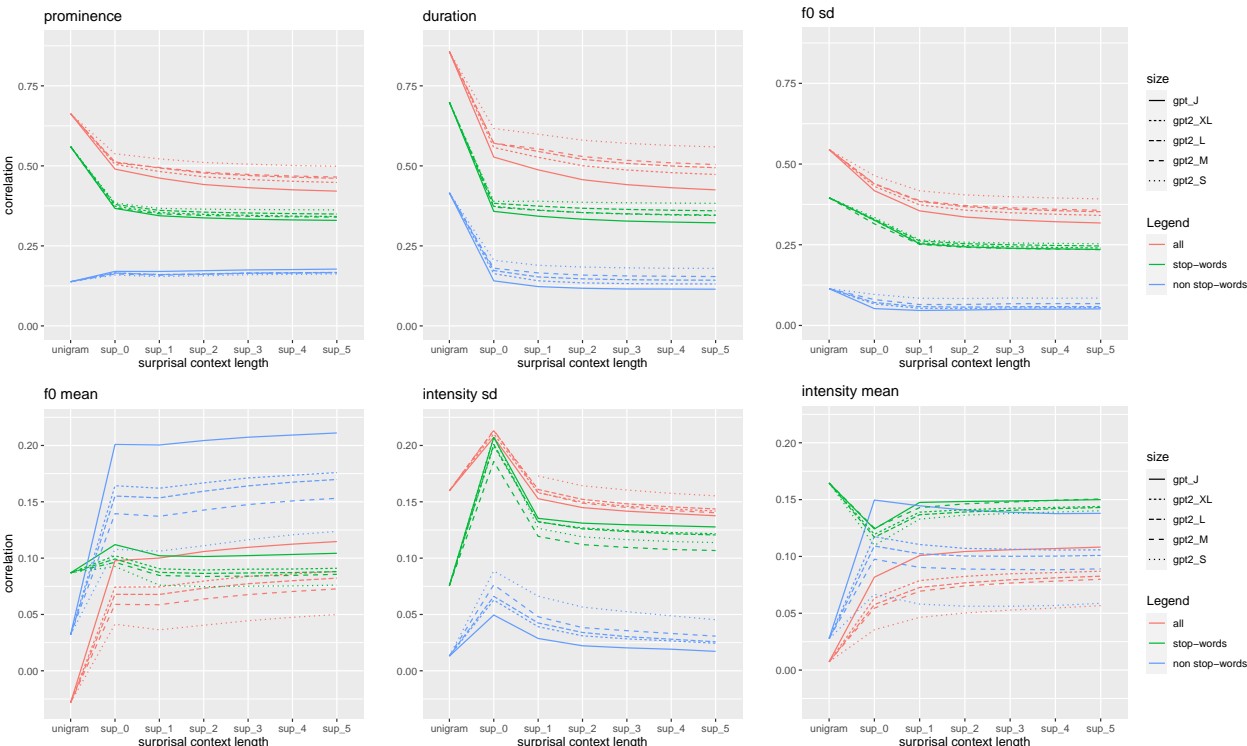

Figure 4: *Spearman's rank correlations between word-level surprisal values and signal-based characteristics for language models of different sizes, for all words in the corpus, for stop-words and for content words. Please note the difference in y-axis scaling between the two rows of plots.*

other measures, they systematically increase with the model size, in particular for the content words. For stop words (except for the largest model) these two measures correlate best with unigram-based surprisal, but for the content words the context provided to the models substantially increases the correlations. This might be partly a consequence of the models' ability to account for the general down-trend in $f_0$ (and, to a lesser degree in intensity) with the words further down the sentence being generally less surprising than those at the beginning. This explanations, however, does not fully explain the differences observed for different model sizes and for $f_0$ also the extend of the provided context.

## 5. Surprisal in Speech synthesis

While the previous analyses showed relatively modest correlations between surprisal values and prosodic features, there is, nevertheless, a possibility that a modern speech synthesizer, as a complex statistical model would be able to find utility for the surprisal values. Two anchor systems were trained; a **baseline** system with no conditioning and a top line system with CWT prominence conditioning (**prom**). To assess the effect of surprisal, we trained two systems conditioned with word surprisal values from the opposite ends of our language model scale; smallest GPT2 with no context (**gpt_small_sup0**) and GPT_J with context of five previous sentences (**gpt_j_sup5**). Based on the correlation analysis, we also added the unigram surprisal value as an input for both systems.

Three last chapters (763 sentences) of the "Report of the President's Commission on the Assassination of President Kennedy" were selected as test material and the rest of the data was shuffled and used as training (11500) and validation set

(333). Each model was trained for 600 epochs.

We synthesized the test material with appropriate conditioning: surprisal values extracted with respective language models with same context sizes as during training. For topline model **prom**, we used prominence labels extracted from the actual speech of the test sentences instead of predicting labels from text.

Initial listening suggested that the differences between the baseline and the surprisal-conditioned systems were fairly subtle, and knowing the difficulties in subjective prosody evaluation, we decided to settle for numerical evaluation of the prosodic features in this work. Instead of extracting the features from synthesized speech, we compared the phone level predictions of duration and $f_0$ from FastPitch. Reference values were extracted by using the trained **baseline** model as an aligner. In this way, the prosody evaluation is not affected by duration differences between different systems or errors in pitch extraction. Root-mean squared error values (in standard units for $f_0$, and 20 ms frames for duration) as well as Pearson correlation with the reference speech are reported in Table 1.

The correlations are generally in line with previous results on this data [25]. We can observe that the surprisal conditioned models are not considerably different from each other. They do provide a small improvement over the **baseline**, though not substantially breaching the gap to the prominence conditioned top line **prom**. If anything, the **gpt_j_sup5** model yields marginally *worse* results than the **gpt_small_sup0** that is using smaller language model and shorter context length. This is in line with the results of the correlation analysis presented in Section 4.2, albeit somewhat surprising for the $f_0$-based measures that showed higher correlations between surprisal values and $f_0$-mean (but not $f_0$-sd) for the larger models.

Table 1: *Results of objective speech synthesis evaluation*

|  | $f_0$ **RMSE** | $f_0$ **cor** | **dur RMSE** | **dur cor** |
|---|---|---|---|---|
| *All words* | | | | |
| baseline | 0.629 | 0.474 | 0.079 | 0.834 |
| gpt_small | 0.620 | 0.489 | 0.078 | 0.836 |
| gpt_j | 0.624 | 0.487 | 0.079 | 0.834 |
| prom | 0.582 | 0.583 | 0.079 | 0.842 |
| *content words* | | | | |
| baseline | 0.630 | 0.492 | 0.067 | 0.871 |
| gpt_small | 0.622 | 0.502 | 0.066 | 0.871 |
| gpt_j | 0.625 | 0.501 | 0.067 | 0.870 |
| prom | 0.583 | 0.595 | 0.067 | 0.873 |
| *stop words* | | | | |
| baseline | 0.629 | 0.417 | 0.082 | 0.779 |
| gpt_small | 0.615 | 0.445 | 0.081 | 0.780 |
| gpt_j | 0.617 | 0.437 | 0.081 | 0.782 |
| prom | 0.580 | 0.544 | 0.081 | 0.793 |

Finally, as the numeric differences between the systems were small, we briefly assessed the controllability of the surprisal-conditioned models, comparing this aspect with the prominence-conditioned model. We synthesized short sentences with manipulated surprisal and prominence values of words in the beginning, middle, and end of sentences. For surprisal models, we manipulated the GPT-based values keeping the unigram values constant, simulating the effect of context. We observed that while the prominence-conditioned model synthesizes variations faithfully (excluding articles and other short function words), the surprisal conditioned models were more erratic in their behaviour, often failing at the phrase boundaries, where the default prosodic pattern such as nuclear stress seems to override the conditioning[4]. Fig. 5 illustrates a typical behaviour.

## 6. Discussion and Conclusions

In this work, we investigated the use of word surprisal, a measure indicating the amount of new information conveyed by a word, as a feature to aid the prosody in speech synthesis. Our primary focus was to investigate surprisal with respect to it's ability of encoding prosodic prominence in human and synthesized speech.

The results of statistical analysis comparing surprisal values, context sizes, acoustic features, and prosody point at possible issues with this approach. The general observation was that prominence correlated with surprisal rather weakly, and increasing the model size and the context window had mostly detrimental effect. The synthesis results also suggest a limited efficacy of using surprisal values for eliciting appropriate prominence patterns, while providing a minimal improvement over the baseline.

LLM-based word predictablility and prominence patterns undoubtedly do correlate. As depicted in the Goldilocks story, (Fig. 1), GPT2 recognizes new and important information, as well as old information given in context. But although our analysis was based on a single speaker, there appears to exist fundamental differences between surprisal values and the human way of highlighting words appropriately in speech. LLMs consistently underestimate the prominence of given words in longer contexts. This is partly due to their better memory, but the words that are repeated are also often worth repeating, and thus uttered prominently.

---

[4]For example, the synthesizer failed in synthesizing the Goldilocks passage shown in Fig. 1, insisting on emphasizing 'bear' each time.

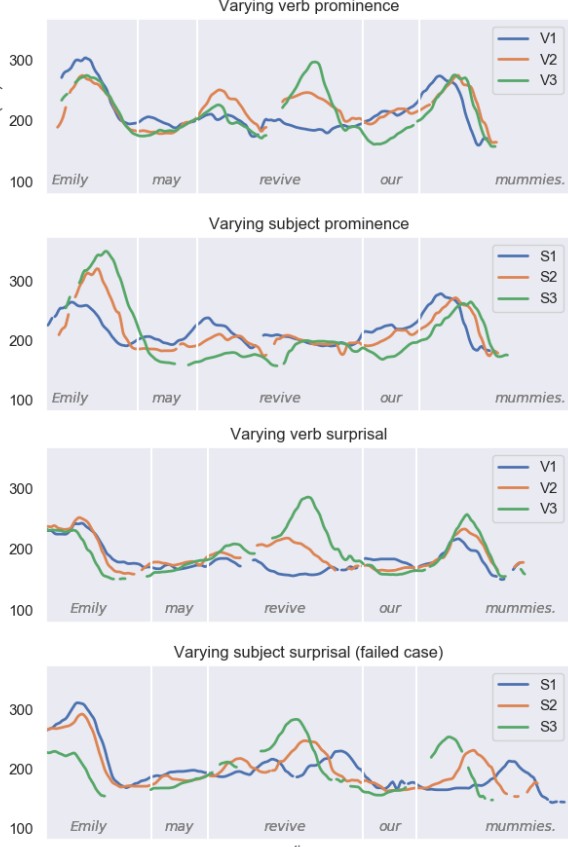

Figure 5: *$f_0$ trajectories of synthesized utterances illustrating the controllability of word prosody by prominence and surprisal conditioned systems. The numbers 1-3 refer to low, average and high prominece and surprisal, respectively.*

There are also issues related to shorter, phrase-sized context. As also noted in [20], the language models naturally tend to assign low surprisal for the phrase endings and first names instead of surnames, contradicting the typical English pattern of assigning stress to the final word. We suspect this is the main reason for the better performance of unigrams, that model the inherent informativeness of a word.

Furthermore, human interaction and context of communication, namely, what is informative and relevant to the audience, plays a significant role in determining prosody, something that the introspective surprisal values can not model. The data exposure of the larger LMs far outweigh the learning ability of a typical human speaker or listener, rendering what is surprising to the LM and human might be very different.

Finally, the information theoretical aspects explain only a part of the prosodic variation. For example, rhythmic constraints and temporal chunking play a large role in prominence placement, in a language dependent way [32].

While our results indicate a limited usefulness of surprisals generated by LLMs for speech synthesis prosody, additional measures derived from the token predictability combined with other LM- based features will likely contribute to better modelling of speech prosody in TTS applications.

## 7. Acknowledgements

S.K. was supported by the Academy of Finland project no. 340125. The authors wish to acknowledge CSC – IT Center for Science, Finland, for providing the computational resources.

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
