# OpenReview forum: "Investigating the Utility of Surprisal from Large Language Models for Speech Synthesis Prosody"
_Interspeech.org/2023/Workshop/SSW — SSW12_

### Official Review · Reviewer_1UXP · 2023-05-30
**Surprisal from Large Language Models: should be accepted**

**Rating:** 9
**Confidence:** 5

**Review:**

Key Strength of the paper: Generally well-written, technically sound, includes an objective evaluation, good summary of
related work, good introduction to the concept of surprisal

Main Weakness of the paper: The discussion of the main finding, viz. that prominence is only weakly correlated with surprisal, is rather thin on theoretical implications (see below).

Novelty/Originality, taking into account the relevance of the work for the SSW audience: Highly relevant. Explicitly accounting for surprisal in modeling TTS prosody is novel (to the best of my knowledge).

Technical Correctness. The work appears to be technically solid. The paper gives sufficient detail to reproduce the analysis.

Suggestions for improvement: The main conclusion is that prominence is only weakly correlated with surprisal. The authors state that "surprisal does not seem to add relevant information in terms of prosody." I believe that they have it backward and are missing an opportunity here. The Smooth Signal Redundancy hypothesis advocated by Alice Turk and Matthew Aylett posits that any effect of information-theoretic factors (in their terminology: language redundancy) is entirely absorbed and modulated by prosody and stands in a complementary relationship with signal redundancy (i.e., the acoustic realization of the message). But recent work has provided evidence that there is an effect of surprisal above and beyond what prosodic structure encodes. The size of this additional effect is small compared with that of prosodic structure (phrase boundaries, accenting for prominence) but non-negligible. And this is exactly what is reported in the present paper! Information-theoretical aspects are not expexted to explain prosodic variation to the full extent, as suggested in the Conclusion, but to exert an effect of their own.

I would also like to point out that surprisal should not be expected to reflect the effect of repeated occurrence of a lexical item on its prominence. First, second (and repeated) occurrence focus does not necessarily result in reduced prosodic prominence and, second, surprisal is more related to local than global (un)predictability. If this is incdeed the case, then capturing large contexts may not be conducive to better predict prominence.

Quality of References: Adequate, except for literature on prosody vs information-theoretic factors (see above).

Clarity of Presentation: Generally clear. However, figures are difficult to read when printed on paper. The lines in fig. 4 are particularly difficult to distinguish.

Note: The paper exceeds the page limit for submissions by a full page.

---

### Official Review · Reviewer_rgUU · 2023-06-04
**The authors investigate how Word Surprisal, estimated by LLM-based word likelihood can help to improve the TTS prosody. They also explore the correlation between this text-driven metric and Word Prominence, derived from a single speaker voice corpus. Although the results do not demonstrate a strong prosody improvement, the analysis and the findings are of great interest for the speech synthesis community**

**Rating:** 7
**Confidence:** 5

**Review:**

The authors investigate the usefulness of Word Surprisal, estimated by a Large Language Model (LLM) word likelihood, for spoken prosody prediction. They also explore the correlation between this metric and Word Prominence, derived from a single-speaker voice corpus. Although the results do not demonstrate a strong correlation, the analysis and the findings are of great interest to the speech synthesis community. The discussions and conclusions seem correct and grounded.

One comment, though, is regarding the correlation of the proposed metric with the word duration. It is not stated how the duration metric was calculated. The overall word length seems to be trivially correlated with the word surprisal (longer words are rarer than shorter ones), so it seems to be more suitable to have the duration normalized by the number of phones in a word.

---

### Decision · Program_Chairs · 2023-06-14

**Decision:**

Accept

**Comment:**

SSW2003 received 45 papers. The acceptance rate is 82%. We are pleased to inform you that your paper has been accepted by the SSW2023 Program Committee. Please read the reviews carefully and submit your camera-ready paper by June 28th. Most reviewers performed a detailed review. Please answer to their questions and consider their comments. Note that camera-ready papers are credited with one extra page to allow authors to consider reviewers’ suggestions. So max 7 pages in total including figures & refs.
The deadline for submitting the revised version (with full non-anonymized authors and refs!) is 28th June.